# Blockchain's Role in Enhancing Quality and Safety and Promoting Sustainability in the Food and Beverage Industry

**Nir Kshetri** 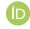

Department of Management, University of North Carolina at Greensboro, Greensboro, NC 27412, USA; nbkshetr@uncg.edu

**Abstract:** The objective of this paper is to assess the potential roles of blockchain technology in enhancing quality, safety, and sustainability throughout the production and distribution of food and beverage products. To achieve this, a multiple case study approach has been selected as the primary research methodology. This article underscores the transformative impact of blockchain implementation on inter-organizational transactions, reducing uncertainty among supply chain participants and fostering more equitable interdependence among partners in the value chain. These developments have the potential to bolster quality, safety, and sustainability within the food and beverage industry. The article also explores strategies for enhancing blockchain's influence on interfirm governance structures within the food and beverage sector. It delves into the possibilities of broadening participation by increasing the number and variety of participants in blockchain networks. It investigates how the synergy between blockchain technology and other emerging technologies can further optimize their impact on reducing interfirm governance structures. Also addressed in the paper is the potential for blockchain-based solutions to enhance distributive fairness within the food and beverage industry, offering marginalized groups, such as small-holder farmers, greater opportunities for integration into the global economy. Special emphasis is placed on blockchain's capacity to enhance interfirm governance in this industry by reducing uncertainty among supply chain participants and creating more symmetrical dependencies among them. The article also posits that by fostering entrepreneurial prospects for marginalized communities and promoting distributive fairness, blockchain technology can contribute to socially responsible actions. Overall, this study extends theories and concepts from information and communications technologies' (ICTs) effects on agency, boundaries, and uncertainty in the context of organizational and inter-organizational dynamics.

**Keywords:** blockchain; food and beverage industry; interfirm relationships; smart contracts; sustainability; uncertainty



## 1. Introduction

Scholars have a keen interest in research that advances our understanding of how to tackle grand challenges [1]. One such pressing challenge we face today is the proliferation of adulterated, deceptively packaged, and counterfeit food products, which pose significant health risks. Annually, approximately 600 million people worldwide fall ill due to contaminated food, resulting in around 420,000 fatalities, including 125,000 children under the age of five [2]. Amid the myriad challenges confronting the food industry, enhancing the sustainability and safety of food products emerges as a pivotal concern. Notably, a study conducted at the household level in the U.S. identified inadequate food quality as a key contributor to food insecurity [3]. The economic toll of food fraud and adulteration exceeds USD 40 billion globally each year, with estimates suggesting that 30 to 40% of consumed food is either adulterated or mislabeled. A survey revealed that 39% of food manufacturers believed their products could be easily counterfeited, and 40% found food fraud challenging to detect using existing methods [4]. Additionally, issues of slavery and forced labor continue to plague this industry [4].

A related grand challenge revolves around the imperative to enhance the sustainability of both the environment and social systems, stemming from the actions of various stakeholders in the food and beverage industry (FBI). Ethical questions have arisen regarding whether smallholder farmers receive fair compensation, as exemplified by coffee producers receiving a mere 2% of the price of a cup of coffee [5]. Moreover, only 10% of this value is estimated to remain in producing countries [6].

The issues of safety and sustainability also intersect with the bottom line for organizations in the FBI. Firms across the supply chain (SC) are increasingly emphasizing food safety and environmental concerns [7]. For instance, a sugar company in Colombia faced demands from multinational corporations (MNCs) to establish explicit corporate social responsibility (CSR) initiatives starting in the 2010s [7]. As this example indicates, companies from developing countries are often forced to comply with powerful MNCs' demands due to what is referred to as asymmetric dependence [8].

Many of these challenges can potentially be addressed through advancements in blockchain technology, recognized as one of the six computing "mega-trends" by the World Economic Forum [9]. Smart contracts, emerging as a key blockchain application, hold the potential to create value for various stakeholders in this industry. Previous studies have investigated the potential benefits arising from the integration of blockchain within the FBI [10]. These advantages encompass the augmentation of food product safety and quality [11] as well as the enhancement of efficiency and transparency [12]. There is, however, little research that examines how blockchain's use in FBI transactions can reduce uncertainty, enhance symmetry among value chain partners, and improve quality, safety, and sustainability. In order to highlight the gaps that this paper attempts to address, let us consider uncertainty in SCs. As mentioned above, MNCs are increasingly demanding developing world-based companies such as sugar companies in Colombia follow CSR standards. Nonetheless, implementing the assessment of sustainability practices among various stakeholders faces challenges due to technical impracticalities [13]. Prior researchers have noted significant gaps between CSR standards and business practices. Such gaps can be attributed to the opaque and complex nature of global SCs [13,14]. Thus, organizations (e.g., MNCs) face ongoing challenges in forecasting the behaviors of other participants (e.g., firms in the developing world) and maintaining control [15]. Uncertainty is thus a key feature of the complex context of global supply networks [15–17]. A key question is thus how blockchain can help address various sources of uncertainty.

This study seeks to address these research gaps by investigating the impact of blockchain on organizational and inter-organizational processes within the FBI. Specifically, we aim to answer the following research questions: (RQ1) How does the deployment of blockchain technology in inter-organizational transactions within the FBI lead to a reduction in uncertainty among SC participants and subsequently enhance quality, safety, and sustainability?; (RQ2) In what ways does the utilization of blockchain technology in inter-organizational transactions create more symmetric dependence among value chain partners, and how does this increased symmetry impact the improvement in quality, safety, and sustainability within the FBI?

Our choice of the FBI as the study's setting is motivated by its significance as a multi-trillion-dollar industry, grounded in the fundamental importance of food in human life and its central role in national economies worldwide [18]. Notably, blockchain applications in this industry have gained prominence, with IBM Food Trust being adopted by major food companies such as Nestle, Unilever, and Walmart. As of mid-2018, this system stored data related to one million items across approximately 50 food categories, facilitating over 350,000 data transactions [19]. IBM officially launched its Food Trust platform in November 2018, and Carrefour partnered with IBM to implement the solution. By 2021, Carrefour was tracking over 30 product lines using blockchain, with plans to expand to 100 product lines by the end of 2022 [20]. As early as 2018, Subway and Tyson were reported to be testing blockchain solutions provided by FoodLogiQ [21]. Research in this domain is poised to

provide insights into the role and limitations of blockchain in addressing critical social and economic challenges within the FBI.

The paper's structure is as follows: We begin by providing an explanation of blockchain and related concepts, followed by a comprehensive literature review. We then proceed to describe our chosen research methods. Next, we establish a framework and put forth propositions concerning the contributions of blockchain technology to uphold quality, safety, and sustainability in the production and distribution of food and beverage products. This is followed by a section on discussion and implications. Finally, we offer concluding remarks.

## 2. Blockchain: Some Background, Concepts, and Facts

In this section, we provide definitions and insights into blockchain and related concepts. Blockchain can be defined as a decentralized ledger that concurrently maintains digital transaction records on multiple computers. In some instances, this involves thousands or even millions of computers distributed across the internet. Once a block of records is added to the ledger, the information within that block becomes mathematically linked to other blocks, forming an immutable chain of records [22]. This mathematical interconnection ensures that the information within a block cannot be altered without modifying all subsequent blocks. Any attempt to tamper with a block would result in a noticeable discrepancy, promptly detected by other participants in the network [23].

Blockchains employ cryptography-based digital signatures to verify identities. Users sign transactions with a "private key", typically a lengthy and randomly generated alphanumeric code. This private key is exceedingly difficult for hackers to guess and is known exclusively to the account holder. Complex algorithms are used to generate a "public key" from private keys, allowing information to be shared securely. Public keys are openly accessible, like a bitcoin wallet address, which any bitcoin user can employ to send payments. However, only the individual possessing the private key can initiate transactions from that account.

The integrity of a user's identity is safeguarded, and blockchain provides the capability to grant limited access to third parties. Furthermore, blockchain inherently includes an audit trail that thoroughly documents the creation, modification, and deletion of records [23]. Within blockchain-based ledgers, there is no necessity for record-keepers to trust one another. Consequently, the risks associated with centralized data storage by a single owner, common in traditional systems, are mitigated in blockchain applications [23].

The key attributes of blockchain—decentralization, immutability, and authentication through cryptography—are poised to serve as powerful tools for enhancing quality, safety, and fairness within the FBI. These attributes are detailed in Table 1.

**Table 1.** Blockchain's key features.

| Feature | Explanation | Some Uses |
|---|---|---|
| Decentralization | Decentralized network of online registries synchronized to track transactions. | Malicious actions can be detected and prevented. Participants verify information themselves. |
| Immutability | Complete documentation of creation, modification, and deletion of records. | Transactions are auditable. Improves transparency (e.g., access to data about food). Not susceptible to theft, damage, corruption, or fraud. |
| Cryptography-based digital signatures to verify identities | Users sign transactions with a "private key": known only to the person who controls the account. | Enables a required level of authentication, which increases confidence. |

### 2.1. Smart Contract

The extent to which contracts can be effectively upheld is directly correlated with the scale of formal sector entrepreneurial activities [24]. Enhanced contract enforce-

ment, facilitated through smart contracts, is poised to be a pivotal mechanism through which blockchain can promote entrepreneurial endeavors. Undoubtedly, one of the most highly anticipated future applications of blockchain technology is the implementation of smart contracts.

The majority of smart-contract solutions designed to facilitate entrepreneurial activities are hinged on the utilization of blockchain technology. In such scenarios, the code governing the smart contract is securely embedded within the blockchain, and each contract is uniquely identified by its designated address. Users initiate transactions by directing them to this specific address, with the blockchain's consensus protocol ensuring the precise execution of the contract.

*2.2. Permissionless and Permissioned Chains*

Within the realm of blockchain technology, we encounter two distinct paradigms: permissionless and permissioned chains. In the realm of permissionless blockchains, exemplified by bitcoin, which operates as an open platform, participation is unrestricted, and anyone can join the network. Conversely, private or permissioned blockchains operate under more stringent controls, necessitating authorization granted by a designated authority. Among these, permissioned blockchains exhibit a particular proficiency in regulating participation, limiting it to a select group of members, and facilitating the structured sharing and management of data among these participants [25,26]. Utilizing permissioned blockchains, real-time data sharing among stakeholders in food SCs is feasible, accompanied by secure transaction processing. Following a consensus-driven transaction's completion, a permanent record is generated and appended to the existing blockchain as a new block.

Illustratively, Walmart embarked on a permissioned blockchain initiative in 2016, conducting trials to monitor pork products in China and track imports to the U.S. from Latin America. The Chinese trial unfolded at a farm operated by Jinluo, situated in the northeastern Chinese city of Lingyi. Jinluo contributed pertinent data concerning the pork products, encompassing essential documents such as farm inspection reports and livestock quarantine certificates [4]. These documents underwent digitization via an industrial personal digital assistant (PDA), a robust smartphone-like device. Subsequently, these data were seamlessly uploaded to Walmart's blockchain platform in real time.

Walmart's systems incorporate blockchain technology to safeguard a spectrum of information encompassing product details, farm particulars, factory information, batch numbers, storage conditions, and shipping specifics. This safeguarding extends to documents associated with farm inspection reports and livestock quarantine certificates. Walmart serves as the custodian of the blockchain housing these records, and the entire system is underpinned by the Hyperledger platform [27]. Consequently, copies of these records are stored and verified by other network participants, known as peers. It is incumbent upon Walmart to configure these peers to participate in the network, with government agencies playing a crucial role among these designated peers [28].

**3. Literature Review**

This section focuses on four themes: (1) Information and communication technologies' (ICTs) effects on agency and boundaries; (2) Institutions in food SCs, power dynamics and dependence; (3) Uncertainty in SCs; and (4) Blockchain's effects on organizational and inter-organizational processes and outcomes. These four themes are interconnected in a dynamic way and through their influence on the FBI's SC. ICTs facilitate data sharing and collaboration while influencing power dynamics and dependence. They also contribute to addressing uncertainty within SCs. Blockchain technology complements ICTs by enhancing transparency and traceability, which has implications for power dynamics, relationships, and uncertainty. Together, these themes play a crucial role in shaping the dynamics of modern supply chain management (SCM).

### 3.1. Information and Communications Technologies' (ICTs) Effects on Agency and Boundaries

Agency and boundaries offer a valuable framework for understanding ICT's impact on organizational relations [29]. ICTs can function as material agents, performing actions independently, thereby altering the boundaries of human actions. They expand entrepreneurial processes beyond temporal and spatial constraints.

Specificity and relationality are crucial ICT characteristics affecting agency and boundaries [30]. Specificity involves control over actions, impacting predictability. Relationality concerns ICT's relationships with other actors, influencing participation in venture creation processes.

ICTs vary in relational capacity; for example, social media platforms engage diverse participants, while some relationships remain shallow, lacking trust for value creation. Transactional trust can address this issue. Overall, ICTs connect diverse actors, enabling resource combination and modification for enhanced value creation [30].

### 3.2. Institutions in Food SCs, Power Dynamics and Dependence

A state of balanced and symmetric dependence between parties establishes a protective framework, offering mutual safeguards and creating a collective incentive to maintain such a relationship [31–33]. In a unilateral dependence relationship, efforts toward building reciprocal, equitable, and two-way interactions are typically limited [33,34]. There is a potential for an expropriation hazard when only one party in a relationship is obliged to make a commitment [31]. This means that the other party can exploit the profits of the first party [35].

The concerns surrounding food safety and CSR within the FBI are significant for several influential stakeholder groups [18]. These issues are therefore both pressing and complex challenges in the food industry. It is worth emphasizing that, due to substantial price reductions and rising input costs, the food industry has been characterized by relatively low profitability. Consequently, food companies may not have the capacity to allocate additional resources to address less direct issues like safety and SC CSR [18].

The "lemons problem," commonly associated with the temptation to compromise quality, is more prevalent in electronic channels, which increase the likelihood of adverse selection, moral hazard, and fraud [36]. Adverse selection arises from information asymmetry, where one party cannot ascertain whether the other party is being truthful. Similarly, moral hazard arises from the difficulty in determining whether the other party is engaging in dishonest or deceptive behavior. In the food industry, a "market for lemon" problem may emerge if relevant parties lack sufficient incentives to conduct due diligence. To mitigate the adverse effects of information asymmetry, consumers often rely on intermediaries such as third-party certification (TPC) agencies.

Regarding power dynamics, stockholders wield considerable influence within organizations. Porter and Kramer [37] noted that, due to heightened pressures to meet stockholder expectations, philanthropic endeavors have been on the decline. Consequently, it can be argued that powerful entities within the FBI may allocate fewer resources to address unethical issues like forced labor and child labor due to these pressures.

### 3.3. Uncertainty in SCs

The concept of uncertainty serves as a foundational element within a substantial body of research focusing on organizational and inter-organizational dynamics [16]. Given the intricate nature of global supply networks, the management of SC uncertainty stands as a central concern for contemporary businesses [15,17]. For instance, as noted above, evaluating sustainability practices faces technical hurdles, revealing substantial gaps between CSR standards and business realities due to the complexity of global SCs [13,14].

Organizations continually grapple with challenges such as inadequate information and comprehension of the SC and its surrounding context, the inability to accurately forecast the behaviors of other SC participants, and a lack of control over the actions of these participants [15]. A prominent issue confronting enterprises is the protracted and sluggish

nature of SCs, necessitating organizational restructuring and the exploration of alternative SC management strategies [38]. As a result, the mounting pressures to reconfigure and devise novel approaches for SC management have been steadily intensifying over time [38].

Inter-organizational institutions encompass a range of established norms, values, conventions, and routines that govern interactions among organizations [39]. These institutions are intimately linked to interfirm governance arrangements [40], which organizations select based on the prevailing levels of uncertainty within the relationship and their degree of interdependence [8]. Firms forge formal or semi-formal affiliations with other organizations to mitigate uncertainty and manage interdependence [41]. These mechanisms may involve enhancing coordination with partners in the value delivery network (VDN) [42].

### 3.4. Blockchain's Effects on Organizational and Inter-Organizational Processes and Outcomes

Insights drawn from research on blockchain deployment in related domains can also offer valuable guidance for comprehending the potential roles of blockchain in enhancing quality, safety, and sustainability within the FBI.

Prior research has demonstrated that blockchain has the capacity to contribute significantly to essential SCM objectives. These objectives encompass aspects related to cost reduction, quality enhancement, accelerated processes, heightened dependability, risk mitigation, sustainability promotion, and increased flexibility [13,25]. Many of these advantages can be attributed to the augmented transparency and accountability engendered by blockchain [25]. Some researchers have explored blockchain's potential to realize these SC and other objectives, particularly within the context of smart contracts. In the context of the FBI's SCs, potential benefits that could result from implementing blockchain include the enhancement of food product safety and quality, as well as increased efficiency and transparency [10–12].

Additionally, research has delved into the implications of blockchain adoption in developing countries. Kshetri and Voas [43] highlighted blockchain's transformative potential in these regions by combating fraud and corruption, thereby fostering entrepreneurial activities among some of the world's most economically disadvantaged populations. Kshetri [44] provided numerous instances of blockchain applications in developing countries, yielding heightened efficiency and reduced transaction costs. Yermack [45] posited that the early adoption of blockchain in developing countries can be attributed to the convergence of three pivotal factors. First, the existing record-keeping systems in these nations are often inadequate and antiquated, creating a void that blockchain technology is well-suited to address. Second, a pronounced public mistrust of regulatory bodies has emerged. Third, the rapid diffusion of modern ICTs, such as smartphones, within these countries has significantly contributed to this trend.

## 4. Methods

We have developed theory through the analysis of multiple case studies, following the approach outlined by Eisenhardt and Graebner [46]. Multiple case studies offer a more robust foundation for theory construction when compared to single case studies [47].

Yin [48] suggests that case studies are epistemologically justifiable when research questions focus on the reasons behind observed phenomena, when behavioral events are not controlled, and when the emphasis is on contemporary events. Other researchers have noted that the case study method is "appropriate and essential where either theory does not yet exist or is unlikely to apply, . . . where theory exists but the environmental context is different . . . or where cause and effect are in doubt or involve time lags" [49]. This study satisfies these criteria since blockchain research in the context of the FBI is at an early stage of theoretical development. By studying the chosen cases, we can address the research questions presented earlier, which involve investigating how different participants in the FBI manage uncertainty and how technology can foster balanced dependence among these stakeholders and advance quality, safety, and sustainability within the FBI.

Following Eisenhardt and Graebner's [46] recommendations, we have forged connections with relevant literature, identified a theoretical gap in the existing body of knowledge, and articulated explicit research questions. Our endeavor is in line with the guidance provided by prior researchers, such as Bansal and Corley [50], who emphasized the significance of our research questions.

Through this process, we have effectively conveyed the theoretical and practical significance of our investigation into the application of blockchain technology in enhancing the quality, safety, and sustainability of food and beverage production and distribution.

### 4.1. Selection of Cases

One perspective posits that the objectives of case selection in a multiple case study design are akin to those in random sampling. From this viewpoint, the chosen cases should be representative of the population, and there should be variability across dimensions of theoretical interest [51]. Unlike random sampling, multiple case study designs require substantive rather than purely statistical considerations to adequately represent a target population [52].

Logistical and financial considerations, along with the feasibility of data collection, also influence the case selection process [53]. Our selection process focused on cases for which sufficient information could be gathered from secondary sources, recognizing that archival data are a recognized data source for case studies [46].

Following Eisenhard's [54] recommendation, we chose ten cases and employed a combination of the extreme case and diverse case methods [51]. We started with the extreme case method and selected cases that represented extreme values in terms of blockchain deployment in the FBI. These cases are considered extreme as they are among the earliest adopters and developers of blockchain, aligning with the notion that best practice models are suitable for case research methodology [54].

Subsequently, we integrated the diverse case method to select specific firms deploying blockchain in the FBI. This approach aimed to achieve maximal variance along relevant dimensions [51], allowing for theoretical reasons such as (contrary) replication, theory extension, and the elimination of alternative explanations [48] to be accommodated. As an example of contrary replication, we incorporated cases involving different methods for assessing crop quality, thus adding a longitudinal dimension to our study. In Eastern Uganda, farmers transport crops to a central location where Nile Breweries officials assess quality and document details within the system [4,13]. On the other hand, Bext360 employs advanced technology, including machine vision, AI, the Internet of Things (IoT), and blockchain, to grade and track coffee beans using a Bextmachine [55]. The diversity in measurement focus areas for blockchain deployment encompasses quality, safety, and sustainability. In order to achieve diversity, we selected cases representing different combinations of major and minor/no focus areas related to these attributes. This approach allowed for a comprehensive representation, as shown in Table 2. It is worth noting that the use of the extreme case method led to the exclusion of cases that did not have at least one major focus area, rendering cell three in Table 1 empty.

### 4.2. Sources and Characteristics of Data

Gottschalk [56] proposed that both the sources of evidence and the evidence itself should be evaluated. Table 3 illustrates the application of several key criteria recommended by Gottschalk [56].

We evaluated the coherence and internal consistency of the data. Following the recommendations of previous researchers [25], we assessed coherence by comparing different data items for the same point in time and the same data items for different points in time. To illustrate this, we provide the following examples:

(a) Bext360. The company commenced its pilot program in November 2017, and during the same month, it partnered with Moyee and the FairChain Foundation to introduce a blockchain-traced coffee called Token. By April 2018, the world witnessed the

first-ever sale of coffee traced using Bext360's solutions. By June 2018, a significant shipment of 60,000 kg of Ethiopian coffee had arrived in Amsterdam.

(b) Walmart. We conducted a comparative analysis of the different stages and procedures involved in the implementation of blockchain technology for the verification and enforcement of sustainability measures. The major milestones, presented in chronological order, are as follows: (1) October 2016: Commencement of testing food safety and traceability protocols in China and the U.S.; (2) February 2017: Successful completion of the pilot programs; (3) May 2017: Public release of the test results; (4) June 2019: Official launch of Walmart's blockchain traceability platform; (5) November 2020: Expansion of the platform to encompass additional product categories.

**Table 2.** The cases selected and their classification in terms of the focus on quality, safety, and sustainability.

| Quality and Safety Sustainability | Major Focus | Minor Focus/No Focus |
|---|---|---|
| Major focus | (1) Bext360 | (4) Banqu Breau Veritas Swiss Coffee Alliance Humaniq |
| Minor focus/No focus | (2) Walmart Ripe.io Alibaba Jd.com Maersk | (3) |

**Table 3.** Applying Gottschalk's criteria for the archival data used in this research.

| Criterion | Explanation | Example |
|---|---|---|
| Time elapsed between events and reporting | Most newspaper articles were published the same day or the next day of the key event. | In April 2018, the world's first blockchain-traced coffee was released by Bext360. The news was released on 16 April 2018 [57]. |
| Openness to corrections | Corrections are incorporated in many of the outlets used in this article. | If an article in fastcompany.com, from which an article has been cited [58], is corrected, the correction is stated after "Correction:" (e.g., https://www.fastcompany.com/90308095/why-you-should-stop-trying-to-achieve-work-life-balance). (accessed on 15 September 2023) |
| Range of knowledge and expertise of the person reporting the events | We used articles written by knowledgeable reporters and journalists. | An article we cited was written by Frank Yiannas [59], the current Deputy Commissioner for Food Policy and Response at the Food and Drug Administration and the ex-vice president of food safety for Walmart. |
| Corroboration from multiple sources | Data and information were triangulated from multiple sources. We also visited the original source, as suggested by Joselyn [60]. | Data and information about Bext360's solutions were compiled from secondary sources [55,61] as well as news released by the company [57]. |

Ensuring the absence of bias, maintaining source credibility, and verifying the trustworthiness of both the source and content of data are crucial considerations. To achieve these objectives, we adopted a rigorous approach by cross-referencing and validating data and information from multiple sources. Rather than relying solely on the websites of the organizations under analysis, we sourced information from respected third-party entities.

Equally vital are the timeliness and currency of the data. To guarantee that the data remained relevant and up to date, we closely monitored the latest news reports related to the selected cases. Additionally, we periodically visited the websites of the pertinent companies to access the most current data and information available.

### 4.3. Pattern Matching Theory and Data

In rigorously conducted case study research, there is a process of "pattern matching", where theory and data align and propositions remain consistent with the selected cases [46]. To illustrate this alignment and demonstrate how the developed framework can be applied to enhance quality, safety, and sustainability in the production and distribution of food and beverage products, we present a visual summary in Table 4 and Figure 1.

**Table 4.** Pattern matching theory and data.

| Proposition | Examples [Case No.] |
| --- | --- |
| Blockchain → Reduction in uncertainty regarding the actions of SC participants ($P_1$) | Breau Veritas<br>Walmart<br>Swiss Coffee Alliance |
| Blockchain → Symmetric dependence ($P_2$) | Bext360<br>Ripe.io |
| Number of participants → Improvements in interfirm governance structures ($P_{3a}$) | Jd.com<br>Swiss Coffee Alliance |
| Variety of roles of the participants → Improvements in interfirm governance structures ($P_{3b}$) | Walmart<br>Swiss Coffee Alliance<br>Maersk |
| Combination with other technologies → Improvements in interfirm governance structures ($P_4$) | Alibaba<br>Bext360 |
| Blockchain → Increasing the likelihood of disadvantaged groups' engagement in entrepreneurial activity ($P_5$) | Humaniq<br>Banqu |
| Blockchain → Improving the outcome of disadvantaged groups' entrepreneurial efforts ($P_6$) | Bext360<br>Swiss Coffee Alliance |

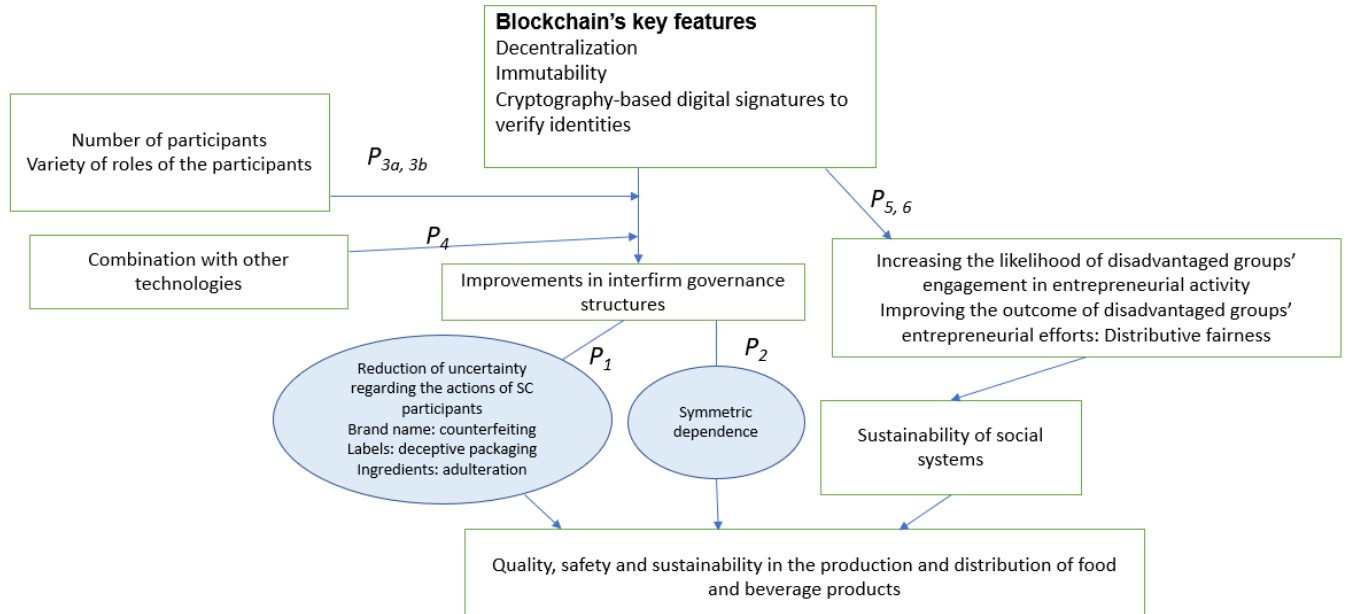

**Figure 1.** Blockchain deployment in the food and beverage industry.

### 5. Blockchain's Roles in Enhancing Quality and Safety and Promoting Fairness in the FBI: A Framework and Some Propositions

*5.1. Improvements in Interfirm Governance Structures*

Table 5 presents blockchain's performance in relation to two pivotal characteristics: specificity and relationality. We focus on how blockchain-based solutions can contribute to the enhancement of interfirm governance by reducing uncertainty and reshaping the dynamics of inter-organizational dependence.

**Table 5.** Key properties of ICTs and interfirm governance: the case of blockchain systems in the food and beverage industry.

|  | Specificity (Value and Effectiveness in Facilitating Transactions) | Relationality (Facilitation of Deep and Meaningful Relationships) |
|---|---|---|
| Uncertainty | Continual verification is possible. Sources of contamination can be pinpointed, and remedial actions can be taken without delay. Smart contract: certainty that the counterparty will fulfill their promises. | Decentralized information flow and chains of immutable records: any alteration of information is likely to be noticed immediately by others in the network. |
| Dependence | Farmers are provided with information to improve certainty about the quality of products (Ripe.io provides information to farmers to improve the quality of produce). Objective measurement: fairer decisions regarding quality. | Powerful supply chain members are likely to face pressures from stakeholders such as regulators and consumers who can also be participants: more symmetric dependence. |

As previously mentioned, food supply networks often exhibit a limited degree of integration and responsiveness, resulting in suboptimal interfirm governance. Blockchain presents a substantial improvement in this regard. For instance, when a retailer asserts that its coffee beans are ethically sourced from a developing country, blockchain technologies, like those employed by the Swiss Coffee Alliance (SCA), offer a considerably higher level of confidence compared to alternative methods. Through blockchain, the entire journey from coffee farm to coffee cup can be meticulously traced, effectively addressing concerns related to misrepresentation.

In a similar fashion, Bureau Veritas, a provider of testing, inspection, and certification services, has designed a blockchain-driven food traceability system catering to consumers. Active participants collaborate in exchanging records and verifying transactions. The system places a strong emphasis on continual verification to provide the highest level of confidence in product history information. The following is proposed:

**Proposition 1 (P₁):** *Blockchain deployment in inter-organizational transactions leads to a reduction in uncertainty in the actions of SC participants, which can help improve quality and safety and facilitate sustainability in the FBI.*

*5.2. Symmetric Dependence*

As mentioned earlier, when significant uncertainty surrounds the outcome of an exchange relationship, opportunistic behaviors are less likely to be detected [62]. In such situations, a participant exploiting a less powerful counterpart faces a reduced risk of reputational harm, potentially leading to the repetition of opportunistic actions in the future [62]. However, when behaviors are observable and opportunistic actions are detected, the incentives for such behaviors diminish. Blockchain systems play a pivotal role in enhancing the detectability of opportunistic behaviors. One of the key advantages of blockchain is its ability to swiftly identify any alterations in a block, triggering corresponding changes in all linked blocks. This creates a noticeable discrepancy that is promptly observed by other participants in the network [23].

The quality of food products exhibits fluctuations [63]. Currently, middlemen have considerable influence in assessing the quality of commodities like coffee and are often motivated to diminish their quality. These intermediaries also hold authority over pricing

and have the discretion to determine compensation for farmers engaged in cultivation [58]. The challenge lies in accurately measuring the quality of commodities, leaving farmers in a state of ambiguity [64] regarding their product's quality. This ambiguity fosters uncertainty about how the quality of commodities, such as coffee, is evaluated [62], making it less likely for unfair behaviors to be detected and more probable for them to recur.

The exploitation of farmers and workers in food SCs is a recognized issue. Some SC partners possess significant power, negating the need for trust. In contrast, less powerful actors, such as smallholder farmers, lack the means to document their value and are thus compelled to depend on their more dominant counterparts.

The variability in food product quality primarily results from a lack of pertinent information sharing among farmers, consumers, and other SC participants. Blockchain enhances the meaningful exchange of information. Companies like Ripe.io use blockchain to aggregate data from farms and various sources to enhance the quality of products like tomatoes. Data collected from farmers and sensors encompass factors such as temperature, humidity, ripeness, color, and flavor of a tomato [63].

Ripe.io collaborated with the restaurant chain Sweetgreen to showcase blockchain's potential in tracking crops. This information is valuable to farmers, food distributors, and restaurants, enabling them to improve the quality of their produce [65]. Such examples underscore how blockchain fosters unique, mutually beneficial, and effective business relationships among value chain partners.

A significant challenge faced by most SCs revolves around asymmetric dependence [8]. Blockchain holds the potential to transform this dynamic by enabling the tracking of small food manufacturers' contributions to value co-creation activities. For instance, as mentioned earlier, solutions like those offered by Bext360 assign a unique ID to each coffee bean, allowing it to be tracked throughout its lifecycle and providing insights into attributes that influence taste. Blockchain can facilitate the establishment of collaborative relationships that are mutually beneficial.

Blockchain deployment can also reduce costs associated with product quality testing, making it particularly effective in terms of the specificity property [30]. The coffee industry, for instance, can incur expenses as high as USD 0.91 per pound of coffee due to paperwork and physical inspections [13]. Blockchain has the potential to diminish the relevance of actors such as certification agencies. Overall, blockchain reshapes the dynamics of dependence. Based on the above discussion, the following proposition is presented:

**Proposition 2 (P₂):** *Blockchain deployment in inter-organizational transactions leads to more symmetric dependence among value chain partners, which can help improve quality and safety and facilitate sustainability in the FBI.*

### 5.3. Number of Participants in a Blockchain Network

Most blockchain systems in the FBI use private or permissioned blockchains. As noted above, such chains are restrictive, and access needs to be granted by some authority. These systems thus vary in terms of the number of participants that are granted access. For instance, consider a blockchain system that has been jointly developed by the Chinese e-commerce giant JD.com and Inner Mongolia's food supplier Kerchin. This system exclusively links these two firms. Kerchin employs barcode scanning to collect and archive data related to its products, which are then integrated into the blockchain. To make any changes to the data, a digital signature is required, and both parties are promptly informed of any modifications or alterations [66].

Other blockchain systems involve a larger number of participants. Take, for instance, the Swiss SCA, which employs Ambrose's sensor-to-blockchain technology to combat unfair profit distribution within global coffee SCs. The participants encompass SCA's extensive network of SC partners, such as farmers, product developers, manufacturers, roasters, and retailers [67]. This innovative approach leverages blockchain, advanced sensors, and smart contracts to create immutable transaction records in the food industry [68].

The elimination of uncertainty necessitates the seamless flow of timely information from diverse sources. Blockchain-based solutions can exert a significant impact, even if only a fraction of participants opt for their implementation. Moreover, the efficacy of blockchain-based solutions is expected to grow in parallel with the expansion of network effects. Several mechanisms can stimulate these network effects as the number of SC participants increases.

Firstly, under certain conditions, participants may be highly motivated to collude and input false information into the blockchain. An increase in the number of participants can reduce the likelihood of collusion among them.

Secondly, within a food SC, a company can be held responsible for issues arising from its suppliers or even its suppliers' suppliers. By integrating them into a blockchain system, transparency and, consequently, accountability can be fostered among all SC participants. Incorporating a larger number of participants into the blockchain can thus ensure responsiveness and accountability.

Thirdly, it is crucial to ensure the accuracy of information entered into a blockchain. In networks with fewer participants, additional processes may be required to verify the correctness of the information. For instance, JD periodically conducts random spot checks at Kerchin's factories to verify the accuracy and validity of the information [66]. The potential of blockchain to serve as a "truth machine" can strengthen with the growing number of participants. While relationships in contexts like social media often remain superficial despite a high degree of relationality [30], participants in a blockchain model develop meaningful relationships that can be mutually beneficial. A key challenge in social media relationships is the lack of trust necessary for coordination and cooperation [69]. However, trust is established through consensus algorithms and transparency within the trustless system of blockchain. The following is thus proposed:

**Proposition 3a (P₃ₐ):** *Blockchain's impact on interfirm governance structures in the FBI can be improved by increasing the number of participants.*

*5.4. Variety of the Roles of the Participants in a Blockchain Network*

When a wide variety of actors participate in a blockchain network, diverse categories of data and information are likely to be created and entered into the blockchain system, which is likely to reduce uncertainty. To illustrate this argument, we will begin by discussing the case of the Danish shipping company Maersk, which tracked a shipment of avocados and roses from East Africa to Europe in 2014. The primary objective of this initiative was to gain insights into the physical processes and paperwork involved in cross-border trades [70]. Typically, containers can be loaded onto a ship in a matter of minutes, but delays often occur in ports due to missing paperwork [71].

A significant source of volatility in SC processes [32,64] arises from the extent to which various SC participants adhere to regulatory requirements. By involving regulatory bodies in a blockchain system, such volatility can be mitigated.

In a Maersk pilot project completed in February 2017, which entailed the transportation of goods from Europe to the U.S., several government agencies were engaged. Participants in this endeavor included the Customs Administration of the Netherlands, the U.S. Department of Homeland Security Science and Technology Directorate, and U.S. Customs and Border Protection [71]. When government agencies are integrated into a blockchain network, relevant paperwork is likely to be seamlessly uploaded to the system. For example, in China, regulators are part of a pilot project initiated by IBM and Walmart aimed at enhancing the transparency of the retailer's supply network by tracing the origins of products such as pork and organic food [72].

In another example, Coca-Cola has formulated a strategy to combat the widespread use of forced labor. They plan to leverage blockchain's validation and digital notary features to establish a secure registry for workers and their contractual agreements. The U.S. Department of State is collaborating in the implementation of this pilot program [4].

Relationality also encompasses the nature of the actors participating in venture creation processes [30]. To shed further light on this concept, we provide a brief description of how blockchain-based systems operate. As previously mentioned, Bext360 integrates blockchain-based solutions with other technologies to assign a unique ID to each coffee bean and track it. In this context, Mainelli [73] has identified three key parties in a typical identity document exchange: (1) the subject of the identity (an individual or an asset, such as a coffee bean), (2) the certifier (e.g., a government agency, an accounting firm, or independent third-party certification (TPC) agencies for organic and fair trade products), and (3) the inquisitor (an entity that makes inquiries regarding the subject for regulatory compliance and other purposes).

Typically, a blockchain transaction involves two separate ledgers [73]. A content ledger contains individually encrypted documents, while a transaction ledger holds encryption key access to document folders that are stored on a series of "key rings". Digitally certified documents pertaining to various attributes are added to the subject's key rings by the certifier, often with the subject's permission. For example, a TPC agency may provide organic certification for coffee beans. Once added to the blockchain, certifiers do not have access to the data. Inquisitors often rely on data verified by a trusted third party [73].

When the subject grants controlled key usage through smart contracts, inquisitors are able to review the documents [73]. With the active participation of these parties, each fulfilling diverse roles, documents stored and distributed via blockchain networks are likely to attain a high degree of authenticity. This collaborative approach can effectively address various sources of volatility in SC processes. Thus, we propose the following:

**Proposition 3b ($P_{3b}$):** *Blockchain's impact on interfirm governance structures in the FBI can be improved by increasing the variety of roles of the participants.*

### 5.5. Combination with Other Technologies

Previous studies have identified a positive correlation between volatility and supplier opportunism in formal contracts, as well as between ambiguity and opportunism in relational contracts [64]. Approaching this from an agency theory perspective [74], ICTs like blockchain serve as material agents capable of performing actions independently of human agents [29,30]. These roles of blockchain can be further enhanced through integration with other technologies.

In terms of the specificity property [30], ICTs play a pivotal role in defining the types of resources different actors can contribute as inputs and how these resources are transformed into outputs. The potential advantages of blockchain's specificity property become more evident when combined with other technologies like advanced QR codes, artificial intelligence (AI), and machine vision. Specifically, these technologies can enhance blockchain's adaptivity by expanding the range of possible actions and interactions [30].

To illustrate, we provide three examples of how blockchain can be synergistically combined with other technologies to exert a more substantial influence on the quality, safety, and sustainability of the production and distribution of food and beverage products.

### 5.5.1. Alibaba

Alibaba has integrated blockchain technology into its domestic SCs. In August 2018, Ant Financial, Alibaba's online payment affiliate, entered into a strategic partnership with the Wuchang municipality in China's Heilongjiang province to track the rice SC [75]. Collaborators in this initiative also include Tmall and Rookie Logistics [70]. A primary objective of this project is to combat the proliferation of counterfeit Wuchang rice, a product celebrated for its superior quality and limited production [76]. Each bag of Wuchang rice available on the Tmall platform is furnished with a QR code featuring a unique identification number. Consumers can conveniently scan this code using a smartphone app prior to making their purchase, enabling them to access information about the specific cultivation field, the seeds and fertilizers used, as well as shipping-related particulars [77].

5.5.2. Solution of Maureen Downey and Everledger to Address Physical Tampering of Wine Bottles

Innovative solutions have emerged that integrate blockchain with advanced technologies to combat unauthorized modifications and physical tampering, particularly with high-value food products. For example, fraudulent practices include emptying expensive wine bottles and refilling them with cheaper alternatives. Counterfeiters have even reverse-engineered systems like Coravin to refill bottles. To address these challenges, wine expert Maureen Downey and Everledger have devised a solution. They implant a small chip beneath a plastic capsule that fits over the wine bottle's existing capsule. If a counterfeiter attempts to pierce the chip, it renders the chip unreadable [78].

A fundamental concept within this perspective is that of bounded rationality. It posits that while individuals engaging in transactions may intend to act rationally, they can only do so within certain limitations. This limitation arises from the fact that human beings have restricted access to knowledge and limited cognitive capacities to process the information available [79].

In many relational contexts, the absence of perceived integrity acts as a barrier to establishing the trust necessary for coordination and cooperation [69]. When decisions are made by machines rather than humans, the need for trust in other SC partners becomes obsolete. In other words, blockchain creates a trust-independent environment within SCs.

In summary, combining blockchain with other advanced technologies enables faster and more resource-efficient actions. Blockchain-based systems can significantly reduce the time required for various actions, making them more efficient than available alternatives in terms of compression mechanisms. Another related mechanism is conservation, which pertains to the reduction in resources required to perform an action [30]. The above leads to the following:

**Proposition 4 (P₄):** *Blockchain's impact on reducing interfirm governance structures in the FBI can be improved by combining this technology with other emerging technologies, such as AI, the IoT, and machine vision.*

*5.6. Increasing the Likelihood of Marginalized Groups' Engagement in Entrepreneurial Activity*

5.6.1. Integration of Small Farmers to the Global Value Chain

As stated earlier, when ICTs, including blockchain, act as material agents, they can alter the boundaries of actions available to human agents [29]. Additionally, ICTs contribute value by enabling actions and transforming the nature of the work that must be carried out [30].

Blockchain can change the boundaries of entrepreneurial activities for small-scale entrepreneurs in developing countries. Many individuals in developing countries are unable to engage in global trade due to the absence of essential requirements like identification documents and bank accounts [80].

Blockchain-based solutions have been launched to address these challenges. The software technology company BanQu's "economic passport" aggregates information from a number of sources, such as financial history, land records, trust networks, and business registrations. Potential borrowers can more easily receive loans by showing such information to potential lenders [81].

BanQu and Anheuser-Busch InBev partnered in 2018 to launch the Chembe cassava online-buying project in Zambia, which uses blockchain technology to improve SC transparency and traceability. Zambian Breweries, a subsidiary of Anheuser-Busch InBev, uses BanQu solutions to track its products from farm to fork. BanQu uses GPS technology to locate farmers, and trained agents use these data to streamline and authenticate transactions. Additionally, geotags, farmer profiles, and other relevant information are securely recorded on the blockchain (https://www.craftbrewingbusiness.com/news/blockchain-breakthrough-poor-zambian-farmers-are-now-empowered-within-ab-inbevs-supply-chain/ (accessed on 15 September 2023)).

This initiative benefits rural farmers who lack or have limited access to banking services. The immutable records of their economic activities tied to their digital profiles enable them to engage with NGOs, local cooperatives, microfinance institutions, and banks for loans, grants, and training opportunities. Thus, the following is proposed:

**Proposition 5 (P5):** *Blockchain-based solutions can improve small-holder farmers' chances of being integrated into the global economy.*

5.6.2. Improving the Outcome of Marginalized Groups' Entrepreneurial Efforts: Distributive Fairness

The present state of global value chains is marked by an inequitable distribution of benefits. As an illustration, consider the vast global coffee industry, estimated at a staggering USD 200 billion. Shockingly, coffee producers are estimated to receive a mere 2% of the price of a cup of coffee, with only 10% of this sum thought to remain in the countries where coffee is produced [6]. The FairChain coffee initiative by Moyee strives to raise this percentage to 50% [57].

The sustainability of the farming system in the developing world is not generally viewed as urgent. When customers have access to information regarding the way farmers are paid, social sustainability may be viewed as urgent and important.

Due to blockchain's transparency and detailed information about how value addition is distributed in the food SC, consumers are likely to feel the urgency of issues related to the exploitation of farmers and farm workers. As noted by Mitchell et al. [82], consumers are likely to be more attentive and responsive to the needs of farmers.

The assessment of a technology's performance can be gauged by its capacity to facilitate transactions [30]. When the objective of a transaction is to achieve distributive fairness, it becomes crucial to examine their roles in rewarding fair conduct and penalizing unfair actions. Previous research has observed that, in certain circumstances, unfair behaviors may incur penalties [83]. In experiments designed to investigate behaviors in the ultimatum game, researchers have revealed that individuals are willing to forgo some monetary gains to sanction unfair behaviors [84–86]. Nevertheless, a challenge often arises due to the absence of data to evaluate the fairness of certain participants' actions.

Blockchain deployment in food SCs is likely to make behaviors more observable and thus ensure a higher degree of fairness among different value chain participants. In such a situation, an actor is less likely to pursue strategies for maximizing income [87].

Blockchain-based solutions assure fair wages across the food value chain, as demonstrated by Denver's coffee roaster, Coda Coffee. Coda Coffee uses blockchain to track coffee from African farms to U.S. coffee shops, using solutions developed by Bext360 that consist of Stellar blockchain, cloud-based software, and smart contracts [55]. Different entities, including farmer cooperatives and Great Lakes Coffee, a coffee exporter based in Uganda, furnish primary data related to product evaluation and payment to coffee growers. In April 2018, Coda Coffee sold what it claimed to be the world's first blockchain-traced coffee [88].

As another example, a Dutch startup, Moyee Coffee, has also adopted this solution, collaborating with the blockchain companies FairChain Foundation and Bext360 to introduce a blockchain-traced coffee product known as Token. By June 2018, blockchain technology was employed to monitor the export of 60,000 kg of coffee from Ethiopia to the Netherlands [4]. These blockchain solutions effectively tracked the exported coffee, providing tangible evidence of fair and living wage payments to the farmers [57].

However, despite the urgency surrounding blockchain companies' efforts to address shareholder needs, they have often neglected the empowerment of less influential SC partners. Given the growing concerns about reduced philanthropy and increased shareholder pressures [37], the adoption of blockchain technology is likely to compel companies to engage in more philanthropic initiatives. Thus, the following is proposed:

**Proposition 6 (P6):** *Blockchain deployment increases the degree of distributive fairness in the FBI.*

## 6. Discussion and Implications

The challenges associated with low-quality food products and counterfeit ingredients are widely acknowledged. Additionally, the issue of inadequate compensation for smallholder farmers is a pressing concern [13]. Despite the significance of these problems, the inherent difficulty in directly assessing the quality of agricultural products leaves farmers susceptible to exploitation by intermediaries and other participants within the SC. Another challenge lies in the absence of a feedback mechanism to assist farmers in enhancing the quality of their produce. Blockchain technology holds the potential to address these issues and mitigate the exploitation of smallholder farmers. Consequently, the FBI offers a highly suitable context for blockchain applications.

Blockchain offers firms a unique opportunity to showcase their commitment to integrating social equity and environmental justice into their SC operations. Consequently, this technology has the potential to catalyze a competitive drive towards elevated social and environmental sustainability standards among companies—a "race to the top", if you will. Through blockchain, a path emerges to contribute positively to society by pursuing ethical objectives such as upholding human rights [13].

The theoretical framework presented in this paper offers an approach to addressing the two research questions previously posed: (RQ1) How does the deployment of blockchain technology in inter-organizational transactions within the FBI lead to a reduction in uncertainty among SC participants and subsequently enhance quality, safety, and sustainability?; (RQ2) In what ways does the utilization of blockchain technology in inter-organizational transactions create more symmetric dependence among value chain partners, and how does this increased symmetry impact the improvement in quality, safety, and sustainability within the FBI?

Regarding RQ1, blockchain's attributes such as auditability, continuous verification possibilities, and decentralized information flow, in conjunction with smart contracts, contribute to reducing various sources of uncertainty associated with entrepreneurial activities within the FBI. Blockchain is also poised to establish a more balanced interdependence among firms. Notable changes include an increased role for small-scale food growers in the value chain, leading to more symmetrical interdependence. Detailed knowledge of the attributes that yield high-quality coffee, for example, empowers coffee growers to adjust their inputs, thereby enhancing interfirm governance structures. To fully leverage blockchain's potential, it must be combined with advanced technologies such as artificial intelligence (AI), machine learning, machine vision, and the IoT to enable more effective utilization and improve its performance in terms of specificity. The relationality aspect of blockchain can be enhanced by expanding the number and diversity of participants. To overcome some limitations and drawbacks of blockchain, a combination of advanced technologies and traditional human observation-based methods may be necessary, as exemplified by JD's random spot checks at Kerchin's factories.

Concerning RQ2, blockchain applications like cryptocurrencies and micrometering capabilities are instrumental in fostering entrepreneurial opportunities for marginalized groups, including smallholder farmers. The transparency and accountability afforded by blockchain technology are also poised to address the prevailing issue of distributive fairness and equity within SCs. Consequently, blockchain-based solutions are likely to promote the sustainability of social systems. Prior research has underscored the role of ICTs, acting as material agents, in performing actions without direct or complete human control. Within the FBI, further advancements in blockchain and complementary technologies may eliminate forced labor and child labor. Equally significant, consumers will have access to information enabling them to ascertain whether smallholder farmers responsible for growing food products have received fair wages. In general, the immutability feature of blockchain allows interested participants to verify the data recorded in the ledger against real-world conditions, ensuring that the ledger's data remain unaltered and accurately represent real-world information. Although direct verification of the absolute truth may be unattainable, the integration of blockchain with complementary technologies empowers

SC participants to approach the closest possible approximation of reality. As digitization advances and concerted efforts are made to tackle various institutional challenges, this technology is poised to bring us nearer to a more accurate representation of the truth [4].

It is also worth noting that while blockchain technology undeniably offers substantial enhancements in SC sustainability, it also gives rise to certain unintended consequences. Notably, the extensive energy consumption associated with blockchain deployment has become a topic of significant contention in ongoing conversations concerning climate change and the broader issue of global warming [89].

This research has a number of implications.

### 6.1. Implication 1: Sustainability of Business Models

Based on the preceding discussion, it becomes evident that blockchain, particularly when integrated with other technologies, can be effectively utilized to carry out a range of tasks with objectivity, equity, and efficiency. Blockchain can also promote the sustainability of business models of organizations in the FBI.

Effectively managing crisis situations is crucial for firms operating in the FBI to maintain profitability. Blockchain-based solutions offer valuable tools for addressing risks associated with emergencies and crises (as depicted in Figure 2). For example, in cases where contaminated food products are detected, retailers like Walmart can swiftly pinpoint the source of the issue and strategically remove affected items without the need for a complete product recall.

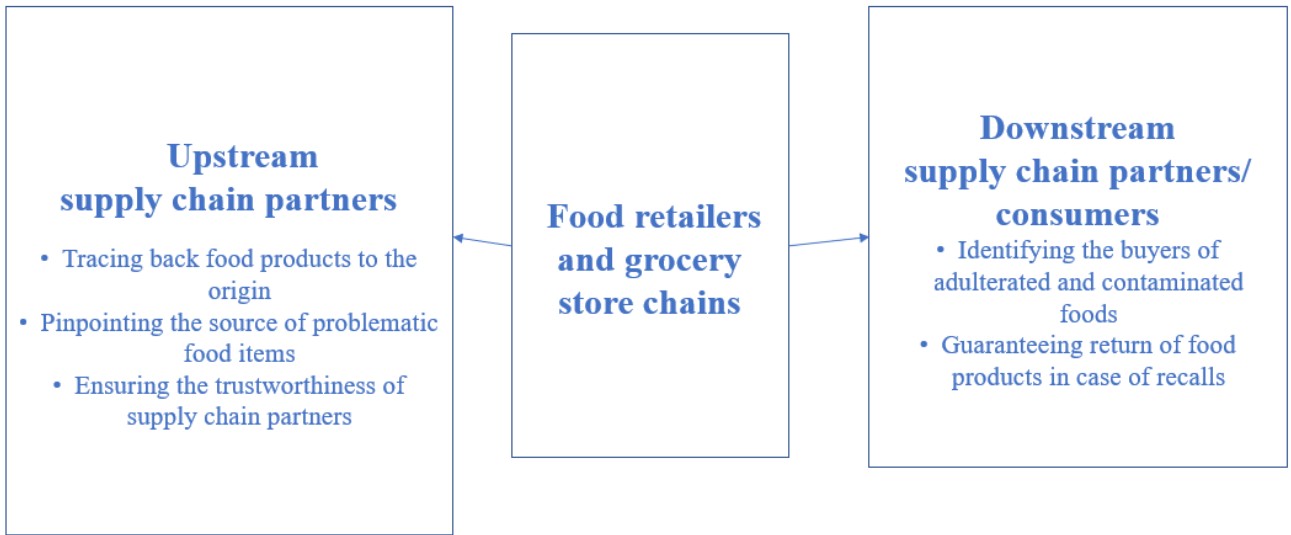

**Figure 2.** Dealing with risk situations: crisis and emergency response.

To illustrate this point, let us consider the *E. coli* outbreak that occurred at Chipotle Mexican Grill outlets in 2015, resulting in 55 customers falling ill. The incident generated negative news coverage, forced restaurant closures, and triggered investigations, leading to significant damage to the company's reputation. Chipotle experienced a sharp decline in sales revenues, and its stock price plummeted by 42%. The root of the problem partly stemmed from Chipotle's reliance on multiple suppliers within its food SCs, which often lack transparency and accountability. Companies like Chipotle faced challenges in real-time supplier monitoring, making it difficult to prevent or efficiently contain contamination once detected [90].

Chipotle's value proposition centered on providing fresh and locally sourced ingredients, yet traditional SC systems that did not utilize blockchain technology were costly and cumbersome. These systems relied on manual verification and extensive record-keeping processes. By leveraging blockchain, the industry could streamline these processes, reducing labor costs and minimizing food waste, thereby achieving substantial benefits [61].

There have been promising advancements aimed at addressing the challenges outlined above. As demonstrated in Walmart's trial of a blockchain-based solution for monitoring pork products in China, blockchain technology facilitates the swift digital tracking of individual pork items within minutes, a significant improvement over the days it previously took. This blockchain system stores comprehensive data, including details about the farm, factory, batch number, storage temperature, and shipping, contributing to enhanced authenticity verification and expiration date validation. Moreover, in the event of food contamination, it offers the capability to precisely identify and recall the impacted products [59].

Although the test focused solely on these two items, it encompassed various stores. If a product is identified as spoiled or the product source is revealed to be compromised, the system takes immediate action. The primary objective is to enhance food safety. The tracked information encompasses details about the farm of origin for vegetables or pigs and their operational methods. These data are collected through commonly utilized technologies, such as RFID tags, sensors, and barcodes, which are already prevalent in numerous SCs [91].

Blockchain has important cost-saving implications for retailers. In the event of a crisis stemming from contaminated food products, retailers such as Walmart can efficiently identify the source and undertake precise removals of affected items, eliminating the need for a comprehensive product line recall. Moreover, blockchain technology empowers a more effective response in cases where tainted products come to light. In this way, the company can keep buyers' confidence in other products and avoid the danger of consumers getting ill [92].

Walmart has detailed its intentions to integrate blockchain technology for various purposes, including verifying the identities of customers and couriers, monitoring container and product temperatures against predefined standards, and other applications. In summary, the adoption of blockchain-based solutions enhances the efficiency of addressing risk scenarios, such as crisis management and emergency response, within the company.

*6.2. Implication 2: The Rank Effect*

Similar to the pattern seen with other technologies, the adoption of blockchain typically starts in larger organizations and gradually spreads to smaller ones, a phenomenon often referred to as the "rank effect" [93]. Presently, most blockchain projects are primarily undertaken by larger companies and are often focused on high-value food products. For example, JD's SC partner Kerchin, which has embraced blockchain technology, reported USD 300 million in revenue in 2017 [66]. Likewise, the French retailer Carrefour initiated a traceability project primarily focused on its premium farm products [21].

Due to the costs and complexities involved, the implementation and management of blockchain systems can be prohibitively expensive, rendering them inaccessible for many organizations. For this reason, blockchain is out of reach for many organizations. For instance, most of China's food supply comes from a large number of small farms.

This stands as a fundamental challenge in achieving food safety within the country [94]. Many of these small farms do not have the resources to embrace a blockchain-based system and supply the requisite information.

Even among big organizations such as Nestle and Gerber, challenges in incorporating blockchain are well recognized. These companies found that moving data from enterprise software such as SAP onto a digital ledger is not an easy task. These companies also need to deal with paper and electronic data in diverse formats produced by farmers, processors, and other SC partners [88].

Naturally, some companies have chosen to confine their blockchain implementation to high-value food commodities. In March 2018, the Chinese e-commerce behemoth JD.com outlined its intention to introduce blockchain, allowing customers to track meat products, initially beginning with top-tier Australian beef [95].

*6.3. Implication 3: The Promotion of Transparency and Accountability*

Consumer interest in the origins of food and beverages is on the rise [96]. Information transparency and ethical conduct are highly valued by consumers. Blockchain technology has the capability to provide consumers with a strong sense of confidence regarding the source and production processes of the food items they enjoy. For example, shoppers can conveniently access a product's history by scanning a QR code before making their purchases, allowing for informed buying decisions.

Blockchain technology can establish a comprehensive framework for transparency and accountability within food SCs. The relationships facilitated by blockchain are likely to carry greater impact and significance. In SCs, where power imbalances and a lack of transparency often persist, influential SC partners can find it challenging to set positive examples, especially when they benefit from the current status quo. Blockchain's transparency has the potential to drive retailers and intermediaries to improve their practices, ultimately benefiting small-scale farmers.

Blockchain is also being used to combat the exploitation of farmers by powerful SC partners, such as retailers. For example, the SCA is developing a blockchain system to securely record data generated by sensors related to the activities of farmers, roasters, product developers, manufacturers, and retailers. The SCA's goal is to use immutable transaction records to prevent retailers from exploiting farmers.

These examples provide compelling evidence that blockchain can address various sustainability-related challenges within inter-organizational relationships. Regulators have also become involved in certain blockchain systems, such as those adopted by Walmart and Coca-Cola. For example, the Walmart Food Trust blockchain system is used to track the movement of food products through the SC, helping to improve food safety and traceability. The Coca-Cola blockchain system is used to track the sourcing of sustainable ingredients, such as cocoa beans and sugar [28]. Overall, blockchain has the potential to transform the food industry by making it more transparent, sustainable, and equitable.

Achieving social responsibility through the equitable distribution of natural capital is crucial. However, it has often been challenging for consumers to ascertain whether farmers and agricultural workers are being justly compensated by food retailers and other influential SC partners. Blockchain technology, with its emphasis on transparency and accountability, offers promising mechanisms to address this issue. It has the potential to compel powerful SC partners to ensure a fair sharing of benefits.

Blockchain introduces unique mechanisms that foster entrepreneurial activities, particularly through the transformation of interfirm governance structures. Notable changes include a shift towards more symmetric interdependence, driven by the increased transparency and accountability inherent in blockchain and smart contracts.

Furthermore, blockchain-based solutions can empower prospective entrepreneurs to engage in entrepreneurial endeavors. Beyond stimulating entrepreneurial activities, blockchain also enhances distributive fairness by promoting equity. It is essential to recognize that the impact of blockchain and smart contracts varies across firms of different sizes and types. Many small farms in developing countries may lack the capacity to fully adopt a blockchain-based system. While blockchain-based IDs and cryptocurrencies play vital roles in facilitating access to global markets for marginalized groups, it is important to acknowledge that blockchain is just one component of the solution for these firms, and these solutions may not entail a fully distributed ledger system.

*6.4. Implication 4: Promotion of the Sustainability of Natural Environment*

Although the primary focus of this paper revolved around blockchain's contributions to bolstering social sustainability, it is important to recognize that this technology can be combined with other emerging technologies to advance the cause of environmental sustainability as well. Prior researchers have demonstrated that, by enabling effective product traceability and continuous monitoring of environmental adherence, blockchain plays a vital role in fostering green SCs [97]. For instance, in the case of Bext360, the

incorporation of sustainable sourcing indicators and the utilization of satellite images to detect potential water pollution by producers demonstrate the potential for a multifaceted approach [98]. The convergence of diverse technologies has the potential to bring us closer to a more comprehensive understanding of sustainable practices and their impact.

**Funding:** This research received no external funding.

**Institutional Review Board Statement:** Not applicable.

**Informed Consent Statement:** Not applicable.

**Data Availability Statement:** No new data were created or analyzed in this study. Data sharing is not applicable to this article.

**Conflicts of Interest:** The author declares no conflict of interest.

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
