# Peer review of "Blockchain’s Role in Enhancing Quality and Safety and Promoting Sustainability in the Food and Beverage Industry"

_sustainability, doi:10.3390/su152316223_

Round 1

Reviewer 1 Report

Comments and Suggestions for Authors

The manuscript is well written addressing a relevant and important theme of blockchain technology and its potential benefits in the food and beverage industry supply chain. The authors have provided evidence of the different ways block chain implementation can positively impact inter-organizational supply chain transactions, enhance fairness in dealings between businesses, and synergies with other technologies for supply chain optimizations. A piece of information that would enhance this manuscript would be inclusion of a short paragraph in the discussion on the potential negative impact on sustainability caused by blockchain technology, especially its significant carbon footprint (for example, reference Biswas et al., 2023, European Journal of Operational Research). It would help readers holistically understand existing guardrails in place while considering blockchain technology and thereby understand the trade-offs between traceability and sustainability for blockchain adoption.

Comments on the Quality of English Language

Quality of English language is good, only some minor editing needed.

Author Response

I am extremely grateful to you for the  detailed, generous and insightful comments on the previous version.  

Reviewer 2 Report

Comments and Suggestions for Authors

This research topic is very interesting. Overall, the research approach of this paper is clear, the reasoning process is logical, and cutting-edge technologies such as blockchain are used to analyze and solve practical problems. At present, there are still some detailed issues in the paper.

(1) Suggest the author to provide additional explanations on the applicability of blockchain technology to this research issue;

(2) It is necessary for the author to further explain the application scope and scenarios of the proposed solution in the conclusion section of the article.

In summary, I suggest accepting this paper with minor modifications.

Author Response

(The authors gave the same response as above.)

Reviewer 3 Report

Comments and Suggestions for Authors

Dear authors and editors:

Please see my referee report which is chosen by the review system.

Comments on the Quality of English Language

Dear authors and editors:

Please see my referee report which is chosen by the review system.

Author Response

(The authors gave the same response as above.)

Round 2

Reviewer 3 Report

Comments and Suggestions for Authors

Dear authors and editors:

The paper shows some improvements compared to the previous version. However, this paper still suffers from nontrivial weaknesses. Please see my referee report which is chosen by the review system.

Comments on the Quality of English Language

Dear authors and editors:

The paper shows some improvements compared to the previous version. However, this paper still suffers from nontrivial weaknesses. Please see my referee report which is chosen by the review system.

Author Response

Response to Reviewer 3’s comments

Reviewer 3 noted a typo in the abstract, different font sizes in the titles of two figures, formatting problems in Implication 4 and commented that there was only one reference to support the implications. All there have been addressed including the addition of the following reference:  

Munir MA, Habib MS, Hussain A, Shahbaz MA, Qamar A, Masood T, Sultan M, Mujtaba MA, Imran S, Hasan M, Akhtar MS, Uzair Ayub HM and Salman CA (2022) Blockchain Adoption for Sustainable Supply Chain Management: Economic, Environmental, and Social Perspectives. Front. Energy Res. 10:899632. doi: 10.3389/fenrg.2022.899632

Round 3

Reviewer 3 Report

Comments and Suggestions for Authors

The paper shows some improvements compared to the previous version. Since all my comments have been basically addressed. However, there are still some spelling errors in the sentences marked in red. Please further examine carefully. I recommend accepting the revised version of the manuscript for publication in Sustainability.

I can highlight it in yellow on the manuscript PDF file. Please see the attachment. For example, words contain quotation marks that make the word incomplete.  This is only my personal suggestion, and only a partial list. Academic papers should be rigorous.  At the same time, I also give recognition to the author's work and paper.

Comments on the Quality of English Language

The paper shows some improvements compared to the previous version. Since all my comments have been basically addressed. However, there are still some spelling errors in the sentences marked in red. Please further examine carefully. I recommend accepting the revised version of the manuscript for publication in Sustainability.

Author Response

Thank you for carefully looking at the manuscript. I have fixed all the typos in the revised paper. Most of them are in the attached file. I found a few more typos that are not in the attached file but only in the final manuscript.   
